# Estimands: bringing clarity and focus to research questions in clinical trials

Timothy Peter Clark ![ORCID],[1] Brennan C Kahan,[1] Alan Phillips,[2] Ian White,[1] James R Carpenter[1,3]

[1]MRC Clinical Trials Unit at UCL, London, UK
[2]Biostatistics, ICON Clinical Research UK Ltd, Marlow, UK
[3]Department of Medical Statistics, London School of Hygiene & Tropical Medicine, London, UK

**Correspondence to**
Dr Timothy Peter Clark; tim.clark@ucl.ac.uk

## ABSTRACT

Precise specification of the research question and associated treatment effect of interest is essential in clinical research, yet recent work shows that they are often incompletely specified. The ICH E9 (R1) Addendum on Estimands and Sensitivity Analysis in Clinical Trials introduces a framework that supports researchers in precisely and transparently specifying the treatment effect they aim to estimate in their clinical trial. In this paper, we present practical examples to demonstrate to all researchers involved in clinical trials how estimands can help them to specify the research question, lead to a better understanding of the treatment effect to be estimated and hence increase the probability of success of the trial.

## BACKGROUND

To illustrate the need for estimands, consider some typical wording for the primary objective of a Parkinson's disease (PD) clinical study:

> The primary objective of this study is to demonstrate that Drug X is superior to placebo in patients with idiopathic Parkinson's disease with 'wearing off' phenomenon.

We are all familiar with such statements, but what do they actually tell us about the treatment effect to be estimated? We know the treatments to be compared, the patient population of interest and that the aim of the study is to show superiority. An experienced researcher in the field would further recognise that these are patients with advanced PD who are currently receiving levodopa and potentially other medications such as dopamine agonists; that these patients have wearing off phenomenon, which means that the effect of levodopa is wearing off between doses, that is, Parkinson's symptoms such as problems with movement (motor symptoms) return; and finally, that the study is being conducted to see if the addition of Drug X to background medication consisting of levodopa plus another anti-Parkinson drug, for example, dopamine agonist or MAO-B inhibitor, will improve control of motor symptoms.[1]

## Postrandomisation events

To fully understand the treatment effect of interest, the objective statement additionally needs to convey what the outcome variable (or endpoint) is, how it will be compared between treatments, when it will be measured and, importantly, how postrandomisation events such as the adjustment of background therapy will be handled in the analysis. The first three attributes are very familiar to many researchers and are not discussed further. However, why do postrandomisation events matter? They matter because adjustment of background medication along with other postrandomisation events such as study medication discontinuation due to an adverse event (AE) can markedly influence the treatment effect being estimated, which in turn impacts the sample size required to detect this effect and ultimately the interpretation of the study results. For instance, in PD, background medication would be changed if the patient is not deriving adequate benefit from study treatment or if the patient is experiencing dyskinesia (involuntary movements) in a clinical trial. The treating physician would normally seek to improve the patient's condition by modifying the dose of levodopa, modifying the time and frequency of administration or adding in a new drug. From this point onwards, the patient is on a different journey from someone who continues on stable background medication, as the benefit they are deriving is affected by the change to background medication.

The researcher must decide in what situation Drug X would be considered *superior* to placebo. Is it when the treatment effect in all patients is considered *regardless* of whether their background medication was changed, or is the focus on the *hypothetical* scenario where no patient had their background medication changed? Or some intermediate scenario? A number of such scenarios are potentially of interest, but as we will see later, can yield quite different treatment effects, of

**Box 1  Estimands in a nutshell**

► Defining an estimand entails specifying five attributes: treatment condition(s), population, outcome, population-level summary and postrandomisation events. In particular it requires careful specification of how postrandomisation events such as treatment modification/discontinuation or use of rescue medication are to be handled.

► The ICH E9 (R1) addendum suggests five strategies to address postrandomisation events:
 – *Treatment policy strategy*: regardless of any postrandomisation events, the treatment effect is described from the final outcome measure in all patients. Note that this approach cannot be used for truncated events, for example, where a variable cannot be measured due to death.
 – *Hypothetical strategy*: the treatment effect under a scenario where the postrandomisation event(s) did not occur.
 – *Composite strategy*: postrandomisation events are incorporated into the outcome definition, for instance, rescued patients are classed as a non-responder for a binary responder/non-responder outcome.
 – *While on treatment strategy*: response to treatment before occurrence of postrandomisation event(s) is of interest.
 – *Principal stratum*: the treatment effect in the 'principal stratum' of interest (eg, the population of patients who would not need rescue).

► Of course, as recognised by ICH E9 (R1), we may well wish to handle different postrandomisation events by different strategies within a study, for example, different strategies for treatment discontinuation due to AE versus lack of efficacy.

differing variability, and hence impact power and sample size. Hence, the importance of clearly specifying the primary clinical question and treatment effect of interest in the protocol, which is where the estimand framework is invaluable.

## The estimand framework

The ICH E9 (R1) Addendum on Estimands and Sensitivity Analysis in Clinical Trials describes a systematic approach to thinking through the trial objectives to ensure that the study goals are both precise and transparent and that the proposed design and analysis is aligned with them.[2 3] The addendum reinforces the importance of a priori defining the *estimand* of interest. This is a precise definition of the treatment effect we would like to estimate: a key part of this definition includes specifying how postrandomisation events (referred to as *intercurrent* events in the guideline) are handled. Based on the chosen estimand, a suitable trial design and analytical approach is chosen to ensure the trial can address the study objectives. The key aspects of the estimand framework and a schematic for common strategies for handling post-randomisation events are given in box 1.

As the ICH E9 (R1) guidance has been discussed in detail elsewhere,[4–12] our focus is on explaining—through contrasting examples—how researchers can effectively use estimands in their trial planning, design and analysis process to ensure clinical questions are precisely specified and the chance of a successful trial is maximised. In so

doing, we hope to convert readers from estimand sceptics to enthusiasts.

We continue with the PD example to illustrate the lack of transparency in current study designs, and how the estimand framework can resolve this. We will explain through a second example in advanced cancer how the estimand framework can be applied in practice.

## EXAMPLE IN ADVANCED PD

To demonstrate the importance of clearly defining a priori *what* the research question is, we use information from two typical phase III trials in patients with idiopathic PD with 'wearing off' phenomenon.[13 14] Both studies have similar design.

During the trial patients receive either experimental or placebo treatment in addition to background therapy. A short titration/adjustment phase is followed by a maintenance phase, during which study treatment is kept stable. The primary efficacy endpoint is the change from baseline to end of double-blind maintenance phase in absolute OFF-time. OFF periods are where PD symptoms (eg, tremor, slowness, stiffness, walking/balance problems) reappear or worsen. The most common postrandomisation event here is adjustment of background medication to control PD symptoms or to manage dyskinesia, a complication of long-term levodopa use.

## What is the estimand?

Neither of the phase III trials in patients with idiopathic PD defined an estimand using ICH E9 (R1) terminology, so we first translated the available information into estimand attributes using the approach proposed by Mitroiu *et al* (which involved a degree of subjective judgement).[15] The primary analysis was performed on what was described as the intention-to-treat (ITT) population, which consisted of all patients who received one dose of medication and had one post baseline OFF-time efficacy outcome assessment during the maintenance phase. To illustrate how estimands work in practice, it will be assumed that endpoints after the first postrandomisation event of change in background medication were not collected, and were instead replaced by the 'last available outcome measure' prior to the change in background medication (ie, last observation carried forward (LOCF)). This approach to the change in background medication and discontinuation of patients is deployed in many PD studies, although it is not explicitly clear that the approach was used in either of the phase III trials.[13 14]

The description of ITT in many PD trials usually implies that the treatment effect to be estimated will correspond to the use of the treatment as introduced into routine clinical practice (ie, the effect of a treatment policy). However, many PD trials actually estimate quite a different treatment effect, which on first glance is not readily apparent:

1. Contrary to the classic definition of ITT, patients who did not receive any study drug are excluded from the

analysis, implying the intended treatment effect is either hypothetical (the effect if all patients *had* received study drug) or principal stratum (the effect in the subset of patients who *would* have received study drug under either treatment), though it is not clear from the description which of these the investigators are interested in.

2. Outcome data after the patient had their background medication adjusted is often set to missing and replaced using LOCF. This implies a hypothetical treatment effect (the treatment effect if adjustment of background medication had not occurred).

In passing, note that we distinguish the estimand from the analysis method used to handle missing data (which should be consistent with the estimand). The estimand enables the researcher to determine what data they need to collect. If this data cannot be collected for whatever reason, then this is a missing data problem to be addressed in the statistical analysis, consistent with the estimand. In this case, using LOCF to impute outcome data after the post-randomisation event is not consistent with the ITT estimand.

In summary, a careful examination of PD trials shows that the treatment effect to be estimated is not precisely and transparently defined. Without this being stated it is not viable to assess whether their design and analysis are appropriate. The studies are often not able (without bringing in untestable assumptions) to obtain a goal of estimating the ITT effect, if this was indeed the intention. The advantage of connecting the study objectives to the estimand upfront in the protocol removes uncertainty about the nature of the treatment effect being estimated, but also focusses attention early in the planning process how to handle postrandomisation events and what consequences this has for study design, conduct and interpretation.

### An alternative estimand

Using the ICH E9 (R1) estimands framework (table 1), we can define a treatment policy estimand, which more closely aligns with the ITT treatment effect, and which requires much weaker assumptions for estimation. Under this treatment policy estimand, OFF-times should be collected on patients after the occurrence of the postrandomisation event, and all data from all patients should be included in the analysis. Table 1 presents the assumptions of this revised treatment policy estimand, together with the typical estimand for PD trials.

### The two estimands compared

Using the information from the phase III trials, we simulated OFF-times to resemble typical studies for PD. To mimic the treatment policy strategy, we assumed that patients with OFF-times above the 90th percentile were poor responders and would have their background medication changed, after which their OFF-times would improve. All OFF-times above the 90th percentile were replaced with randomly selected values from a normal distribution reflecting a treatment from which the patient was likely to derive benefit.

Five hundred patients were randomised in the simulation to experimental or placebo treatment. Thirty-nine patients in the placebo and 11 patients in the experimental arm required change of background medication, because their OFF-time went above the 90th percentile. The table 2 below shows the results of the simulation.

Table 2 shows the treatment effect for the treatment policy estimand (−0.87 (−1.11 to −0.64)) is markedly smaller than for the typical PD trial estimand (−1.14 (−1.43 to −0.86)), although both are statistically significant (p<0.001). Under treatment policy more placebo (39) than experimental (11) patients benefited from adjustment of background medication. Thus, the treatment effect being estimated changes from one strictly comparing the experimental treatment against placebo with no change to background medication in either group to one including changes to background medication in either group (realising that changes may be more common in the less effective group). Subsequently the

| Table 1 | Two estimand strategies for PD trials | |
|---|---|---|
| **Attribute** | **Typical PD trial estimand** | **Alternative treatment policy estimand** |
| Treatment | Experimental treatment or placebo added to background therapy with adjustment of the background medication permitted during the first titration period, but not thereafter | Experimental treatment or placebo added to background therapy, dosed as required, and alternative medication, dosed as required |
| Population | Idiopathic PD with 'wearing off' phenomenon | Idiopathic PD with 'wearing off' phenomenon |
| Outcome | Change from baseline in absolute OFF-time up to the end of maintenance phase | Change from baseline in absolute OFF-time up to the end of maintenance phase |
| Population-level summary | Mean difference | Mean difference |
| Postrandomisation event | Adjustment of PD medication during the maintenance period\nHypothetical strategy; that is, the treatment effect in the hypothetical case where adjustment to PD medication did not occur | Adjustment of PD medication during the maintenance period\nTreatment policy strategy where data after the post randomisation events is collected and used in the analysis. |

PD, Parkinson's disease.

**Table 2** PD simulation results

| | Typical PD trial estimand | | Alternative treatment policy estimand | |
|---|---|---|---|---|
| | **PBO** | **Experimental** | **PBO** | **Experimental** |
| Primary endpoint OFF-time (hours), n | 250 | 250 | 250 | 250 |
| Mean (SE) | 5.28 (0.10) | 4.14 (0.10) | 4.90 (0.08) | 4.03 (0.08) |
| Mean difference (SE) experimental vs PBO | −1.14 (0.14) | | −0.87 (0.12) | |
| 95% CI | (−1.43 to −0.86) | | (−1.11 to −0.64) | |
| P value | <0.001 | | <0.001 | |

CI, confidence interval; n, sample size; PBO, placebo; PD, Parkinson's disease; SE, standard error.

original analysis therefore gives an inflated answer to the treatment policy question.

## WHICH STRATEGY FOR POSTRANDOMISATION EVENTS SHOULD BE CHOSEN?

The strategy depends on the main clinical question of interest. Do we want the best estimate of the effect of experimental treatment, uninfluenced by changes to background medication? This is reasonable, but should not be confused with the treatment policy question. Since typically more patients on the placebo arm are likely to require a change to background medication, for the treatment policy question the comparator is not 'placebo', but a mixture of placebo and change in background medication. Or we may be more interested in something else? Stakeholders will have different views. What is clearly important is that the researcher describes the treatment effect(s) they wish to estimate in a clear and transparent way in protocols and trial reports, in particular ensuring that postrandomisation events are handled consistently with this. Then stakeholders can easily and reliably interpret the trial's findings—the ultimate goal of using the estimands framework.

We now review a number of strategies described in the ICH E9 (R1) estimands framework using the PD example.

### Treatment policy

Under treatment policy, the objective is to determine the effectiveness of Drug X when used as part of an existing treatment algorithm. In this respect it can be seen as pragmatic, as we are comparing treatments under the conditions in which they would be used in practice.[16 17] For instance, in the Parkinson's example, this would mean allowing clinicians to change background therapy during the study. Treatment would be specified as Drug X added to background therapy Y, dosed as required, and with additional medication, as required.

The main implication of this approach for the trial's design and conduct is that all postrandomisation data are collected, which are subsequently used in the statistical analysis. That is, all patients need to be followed up for outcome regardless of postrandomisation events. The other main implication is that the treatment effect may be smaller due to the occurrence of postrandomisation events, which may increase the required sample size. In the case of the PD example, the reason for this is the comparator arm is not a true placebo but placebo plus change in background medication for those patients who do not perform well.

### Hypothetical situation where postrandomisation event does not occur

The clinical question of interest is, what is the effect of Drug X where adjustment to background medication does not occur, regardless of any deterioration of the patient. This is akin to an explanatory trial, that is, a clinical study undertaken in an idealised setting.[16 17] This strategy seems to be favoured by regulatory agencies in studies with PD, as they want to distinguish between the effect of Drug X and an increased efficacy due to adjustment of levodopa.[18]

The analysis methods aligned to this estimand though often require strong assumptions about what would have happened if patients had actually continued treatment uninterrupted. Such assumptions are often unverifiable. For instance, in the PD example, the assumption was that the OFF-time would remain unchanged until the end of the observation period, which is very unlikely. Although this approach is still accepted by regulatory agencies in some clinical settings, it remains controversial.[19]

### Other approaches?

One other possibility is the composite outcome strategy, which involves redefining the outcome to include the postrandomisation event as part of the outcome. For instance, an OFF-time responder endpoint could be defined, for example, 1 hour or more reduction in absolute OFF-time from baseline to endpoint. Patients whose background medication changed could be classified as non-responders.

Including the postrandomisation event in the outcome definition greatly simplifies matters. However, changing from a continuous measure (change in OFF-time) to a binary measure (reduction in OFF-time of 1 hour from baseline) alters the study question being addressed, and

will increase the sample size required to address the question. Further, it is critical to precisely define 'non-responder' so that we do not fundamentally change the clinical question we are asking, even though the treatment effect of interest does change; for instance, if 'non-responder' is broadly defined then this would include symptom control and tolerability as well as changes to background medication. Hence, instead of looking solely at the effect of treatment on symptom control, we could also be looking at symptom control and comparative tolerability. Again, if this is the clinical question of interest, then it is fine, though that would require careful consideration by relevant stakeholders. Critically, it is essential that the target of estimation for a particular trial objective aligns with a relevant clinical research question and that this is clearly defined so that stakeholders can judge its merits: a useful composite must have clear clinical relevance.

## REAL-LIFE SETTING

The above example has been kept simple to illustrate the basic concept of estimands. However, in real life more than one type of postrandomisation event will occur in a study, for example, use of rescue medication, change of background medication, switching from placebo to experimental, study drug discontinuation due to an AE, and so on.

In PD, background therapy can be changed for many reasons, most commonly motor fluctuations (wearing off) and motor complications (dyskinesia). Should we use the same or different strategies to deal with these events? In the first case background medication is adjusted to manage symptoms of the disease, which we are measuring with our outcome variable. In the second case, it is adjusted to manage an AE. We could therefore envisage a hypothetical strategy for motor fluctuations and a treatment policy strategy for motor complications.

Sometimes it may be useful to define multiple estimands within a clinical trial, if different treatment effects are of interest. If so, then researchers must specify, which estimand is the primary and which are secondary.

## CASE STUDY: TRIAL IN ADVANCED CANCER

To illustrate the application of the estimand framework in practice, consider a typical trial in advanced cancer.

### Study objective

In many oncology studies, the protocol has a vague objectives statement such as 'The primary objective of the study is to compare overall survival of subjects treated with experimental therapy relative to those treated with protocol-specific physician's choice'. However, what does this mean in terms of the treatment effect to be estimated?

### Outline study design

Patients are typically randomised to receive experimental therapy plus standard of care or placebo plus standard of care. A common design choice includes allowing placebo patients switching to experimental therapy after disease progression, even though this would not typically occur in practice. In reality patients would switch to another therapy, whenever possible. In the trial setting, patients are then followed up until death or some prespecified time when a predetermined number of deaths have occurred.

### TransCelerate protocol template

TransCelerate BioPharma is an organisation with a mission to drive the efficient, effective and high-quality delivery of new medicines. One area of focus was the development of a standard template for protocols, which included estimands.[20] In order to precisely understand the treatment effect being estimated we propose that the TransCelerate protocol template pertaining to estimands is adopted.

For simplicity, it has been assumed that there is a single postrandomisation event of switching to experimental treatment after disease progression. Post switching data are typically collected and included in the analysis as per the treatment policy strategy, but what does this mean in terms of the treatment effect being estimated and can causal effects be determined? After switching, placebo patients can be assigned to experimental therapy at the discretion of the local investigator, so the treatment effect

| Objective | Treatment policy estimand | Alternative hypothetical estimand |
|---|---|---|
| **Primary** | | |
| The primary objective of the study is to compare overall survival of subjects treated with experimental therapy relative to those treated with protocol-specific physician's choice | *Treatment:* Experimental/Placebo+Experimental on progression at discretion of local investigator<br>*Population:* 'Advanced cancer'<br>*Variable:* Time to death<br>*Population summary level*: HR<br>*Post Randomisation Event*<br>▶ Treatment switching for patients who progress in the control group, to be handled using a treatment policy strategy | *Treatment:* Experimental/Placebo<br>*Population:* 'Advanced cancer'<br>*Variable:* Time to death<br>*Population summary level:* HR<br>*Post Randomisation Event*<br>▶ Treatment switching for patients who progress in the control group to be handled by a hypothetical strategy (ie, the treatment effect if participants did not switch to experimental after progression) |

**Table 3** TransCelerate estimand template

HR, hazard ratio.

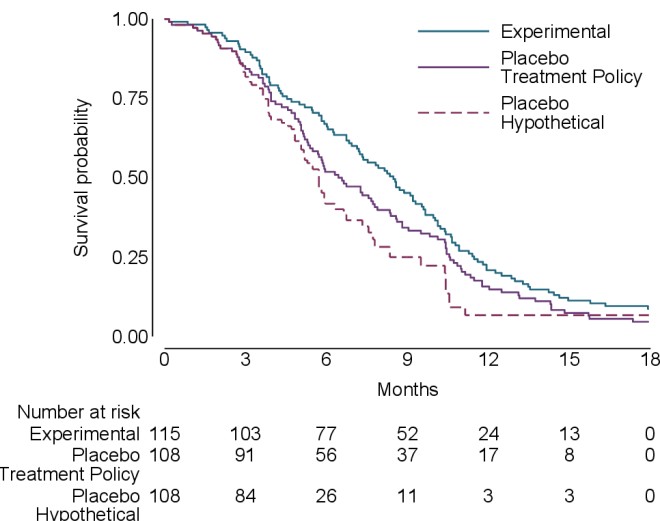

**Figure 1** Kaplan-Meier survival plots for oncology study.

being estimated equates to experimental therapy versus placebo combined with experimental therapy for many patients.

An alternative approach which leads to a more direct causal effect would be to consider a hypothetical strategy for the postrandomisation event of switching. Table 3 below summarises the treatment policy and the alternative hypothetical estimand.

The alternative estimation method of analysis aligned to the hypothetical estimand is inverse probability of censoring weighting (IPCW).[21] IPCW allows us to assess the clinical benefit of the experimental arm compared with the control, by recreating an unbiased scenario where no placebo patient switched to experimental treatment.[22]

### What is the impact?
To illustrate the different estimands under consideration, data were generated to resemble a trial comparing an experimental anticancer treatment with placebo in advanced cancer akin to Latimer *et al.*[23] The data comprised the randomised group, time of switching, time of event/censoring and event indicator. Two-hundred and twenty-three patients were randomised (115 to experimental, 108 to placebo) with 49 switching from placebo to experimental treatment. The results (figure 1) show that the treatment policy and hypothetical survival curves are very similar for the first 3 months and then diverge,

with the hypothetical curve showing greater benefit. As a result, the estimated hazard ratio (HR) changes from 0.79 for the treatment policy estimand to 0.62 for the hypothetical estimand (table 4). The hypothetical estimand is statistically significant whereas the treatment policy estimand is not. This likely reflects the fact that a high proportion of placebo patients switched to experimental treatment (49/108 [(45%)).

## WHICH STRATEGY SHOULD BE CHOSEN?
The actual estimand of interest varies by stakeholder. It can be argued that the hypothetical strategy provides a better causal estimate of the treatment effect. However, this estimate does depend on the methodology used and the underlying untestable assumptions. For instance, IPCW requires measurement of prognostic factors that predict treatment switching and can be prone to error when switching proportions are high.[22] The treatment policy strategy on the other hand has the advantage of including follow-up data on all patients in the analysis, which may be more directly relevant to patients— although interpretation of this comparison depends on how closely study treatment reflects clinical practice.[24] Furthermore, a high switching proportion reduces the effect size, meaning a larger sample size may be needed under the treatment policy strategy.

Regardless of the strategy used, it is imperative to a priori precisely and transparently define the treatment effect to be estimated via estimands. The TransCelerate protocol template approach now includes the estimand attributes, which has resulted in greater transparency and clarity regarding study objectives. As illustrated, clinical trial results can be simulated to illustrate the impact of the different options for post randomisation events to stakeholders.

## DISCUSSION
The examples presented—although relatively straightforward—demonstrate that both (1) using the estimands framework brings much-needed clarity and transparency to the design and analysis of clinical trials, and (2) the choice of estimand has a clinically relevant impact on the trial results. Using estimands results in clearer answers to well specified questions, and so in turn improves regulatory and clinical

| Table 4 | Oncology study results | | | |
|---|---|---|---|---|
| | **Treatment policy estimand** | | **Hypothetical estimand** | |
| | **Placebo** | **Experimental** | **Placebo** | **Experimental** |
| Number of patients | 115 | 108 | 115 | 108 |
| Number who switch | 49 | | 49 | |
| HR | 0.79 | | 0.62 | |
| 95% CI | (0.60 to 1.04) | | (0.43 to 0.88) | |
| P value | >0.05 | | <0.05 | |

CI, confidence interval; HR, hazard ratio.

decision-making. Estimands also facilitate meta-analysis, potentially alleviating an important source of heterogeneity.

In the light of this, we now use estimands routinely when planning our studies. This gives invaluable guidance to choosing which data need to be collected to answer the clinical question of interest. Inevitably this entails planning for a number of different postrandomisation events. Because handling of postrandomisation events in the analysis can require untestable assumptions, it is important to plan for and perform sensitivity analyses (consistent with the estimand) to test the robustness of the conclusions to other clinical plausible assumptions.

In conclusion, estimands are not just a talking point for statisticians. Rather, they are a key practical tool for the whole trial team, to ensure clinical questions, study design and analysis remain focused and aligned. Following our own practice, we therefore advocate their use for all trials, and argue that they should be a mandatory part of published trial protocols.

**Acknowledgements** We thank Nicholas R Latimer for help in creating the second simulation study.

**Contributors** TPC, JRC, AP and BCK initiated the project. The first draft of the manuscript was written by TPC and AP and was refined by JRC, IW and BCK. TPC and IW conducted the simulations. All authors contributed to the manuscript and read and approved the final manuscript.

**Funding** JRC, BK and IW are supported by MRC programme grants MC_UU_00004/07 and MC_UU_00004/09.

**Competing interests** We have read and understood the BMJ Group policy on declaration of interests and declare the following interests: TPC and AP are employed by the clinical research organisation ICON.

**Patient consent for publication** Not required.

**Ethics approval** This study does not involve human participants.

**Provenance and peer review** Not commissioned; externally peer reviewed.

**ORCID iD**
Timothy Peter Clark http://orcid.org/0000-0002-3783-6683

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
