## [Reviewer comments · BMJ Open]

ARTICLE DETAILS

TITLE (PROVISIONAL)	Estimands: Bringing clarity and focus to research questions in clinical trials
AUTHORS	Clark, Timothy; Kahan, Brennan; Phillips, Alan; White, Ian; Carpenter, James

VERSION 1 – REVIEW

REVIEWER	Delvaux , Nicolas KU Leuven
REVIEW RETURNED	15-Sep-2021

GENERAL COMMENTS	This paper illustrates and elaborates on guidance provided by the ICH E9 Addendum on Estimands and Sensitivity Analysis in Clinical Trials. It describes 2 clear cases where alternative choices regarding estimands and post-randomisation events could have greatly altered the measured effect in a clinical trials. However the choices made during the design of the trials were not transparent. 5 strategies to address post-randomisation events are reported and then explored in the subsequent examples. The examples are clear and sufficiently illustrate the challenge at hand. I have only one major comment that could be considered: 1. the cases presented are usefull to illustrate the alternative strategies, however the format used is somewhat confusing. Case 1 starts by comparing the strategy used with an alternative (in quite some detail). Later the alternative strategies are discussed, albeit outside the context of the PD case. It may be more informative to use both cases to illustrate how different strategies can provide very different results and in a third section outline the 5 different strategies and guidance on when and how to use these strategies. There are some minor edits that can be considered to improve readability: 1. line 109: the stament talks starts with "neither study", but considers only a single study on PD symptoms. It is unclear what the second study is being referenced?2. line 203-205: starting this topic with a question and subsequently answering it without actually providing an answer ("it depends") does not read well.
--

REVIEWER	Zwarenstein, Merrick Schulich School of Medicine and Dentistry, Epidemiology and Biostatistics
REVIEW RETURNED	23-Sep-2021

GENERAL COMMENTS	Hi,
-----

	I enjoyed reading this paper. I have two suggestions- first, in lines 122-125, I think a sentence is missing the word "not", (which is present in the related table). The text follows, with the NOT added in CAPS. Contrary to the classic definition of ITT, patients who did not receive any study drug are excluded from the analysis, implying the intended treatment effect is either hypothetical (the effect if all patients had received study drug) or principal stratum (the effect in the subset of patients who would have received study drug under either treatment), though it is not clear from the description which of these the investigators are interested in. Outcome data after the patient had their background medication adjusted is often set to missing and replaced using LOCF. This implies a hypothetical treatment effect (the treatment effect as if adjustment of background medication had NOT occurred). This second suggestion is entirely at the authors discretion and I also want to point out my interested status in this. I think that there is a connection between the clarity which he estimates framework brings to specification of outcomes and also to issues that would usually be thought of as "ITT or some other analysis". I think this clarity is relevant to the framework on pragmatic and explanatory trials, as proposed by Schwartz and Lellouch in 1967 and since incorporated into he PRECIS and PRECIS 2 tools, of which I am an author. My suggestion is that the authors consider pointing out the overlap between the choice of a "treatment policy outcome" as per ICH E(9) with the pragmatic approach proposed by Schwartz and Lellouch for trials where the purpose is to support real world clinical or policy decisionmaking; and, on the other end of the spectrum, the overlap between the "Hypothetical strategy" as per ICH E9 R with the Explanatory approach suggested by S&L for trials whose research question focusses on confirming (or not), a causal hypothesis on the mechanism of action.
--	---

REVIEWER	Cobo, Erik Universitat Politecnica Catalunya, Statistics and Operational Research
REVIEW RETURNED	26-Sep-2021

GENERAL COMMENTS	I have carefully read the comment on "Estimates ..." and I think this is an important article to highlight how important it is to design a clinical trial with a single question, which is not affected by post-treatment decisions. However, I remain skeptical that estimands could "reverse" this difficult situation. Let me apologize for my poor English. Major comments I regret to say that I remain skeptical about the convenience of designing a pivotal trial based on these methods. Of course, I remain convinced that they provide a lot of clarity in the very difficult situation in which, due or to poor design or execution of the clinical trial, the treatment effect cannot be easily interpreted. By moving away from its experimental design, it is closer to the observational one and all interpretations are tentative and therefore cannot offer guarantees to future patients.
---

	In my opinion, estimands help to clarify how complicated it is to interpret a clinical trial in which, for some reason, decision-making is authorized once treatment has started, making untenable the assumption of a similar effect on which the authorization of the product rests for all patients within the eligibility criteria Minor Comments The authors speak of statistical significance 5 years later of the ASA statement. I interpret that only the P-value obtained from the main analysis defined in the protocol can be interpreted as a Neyman-Pearson test. The other P-values, without a priori power, should be eliminated.
--	---

VERSION 1 – AUTHOR RESPONSE

Reviewer	Reviewer Comments	Author Responses
1	The cases presented are useful to illustrate the alternative strategies, however the format used is somewhat confusing. Case 1 starts by comparing the strategy used with an alternative (in quite some detail). Later the alternative strategies are discussed, albeit outside the context of the PD case. It may be more informative to use both cases to illustrate how different strategies can provide very different results and in a third section outline the 5 different strategies and guidance on when and how to use these strategies.	The two examples have different aims. The PD example is intended to illustrate how the lack of transparency in current study designs and how results change dependent on the estimand used. This the reason why the comparison is quite detailed, as we want to demonstrate to researchers why the framework is important. We then go on to illustrate the pros and cons of the strategies proposed in the ICH guidance using the PD example. We have made that clearer in the text. The second example illustrates how the estimand framework can be applied in the planning stage, in particular how simulation can be used to estimate the effect of the estimand on the study outcome. Again, we have made this clearer in the text. Finally, post randomisation/intercurrent events are dependent on the specific details of each study and its design. All of the strategies are potentially relevant for each post randomisation/intercurrent event. The applicable post randomisation/intercurrent events depend on the disease area and the specific clinical question and the pros and cons of applying each strategy depend on the clinical question being asked. Subsequently specific guidance is not practical, although we have discussed the use of specific post randomisation/events and relevant strategies for the PD and oncology studies discussed.
1	Line 109: the statement talks starts with "neither study", but considers only a single study on PD symptoms. It is	We based the analyses on two PD studies, see lines 96-97. The text has been revised for clarity.

	unclear what the second study is being referenced?	
1	Line 203-205: starting this topic with a question and subsequently answering it without actually providing an answer ("it depends") does not read well.	The text has been revised for clarity.
2	I have two suggestions- first, in lines 122-125, I think a sentence is missing the word "not", (which is present in the related table). The text follows, with the NOT added in CAPS. Contrary to the classic definition of ITT, patients who did not receive any study drug are excluded from the analysis, implying the intended treatment effect is either hypothetical (the effect if all patients had received study drug) or principal stratum (the effect in the subset of patients who would have received study drug under either treatment), though it is not clear from the description which of these the investigators are interested in. Outcome data after the patient had their background medication adjusted is often set to missing and replaced using LOCF. This implies a hypothetical treatment effect (the treatment effect as if adjustment of background medication had NOT occurred).	The text has been revised as requested.
2	This second suggestion is entirely at the authors discretion and I also want to point out my interested status in this. I think that there is a connection between the clarity which he estimates framework brings to specification of outcomes and also to issues that would usually be thought of as "ITT or some other analysis". I think this clarity is relevant to the framework on pragmatic and explanatory trials, as proposed by Schwartz and Lellouch in 1967 and since incorporated into he PRECIS and PRECIS 2 tools, of which I am an author. My suggestion is that the authors consider pointing out the overlap between the choice of a "treatment policy outcome" as per ICH E(9) with the pragmatic approach proposed by Schwartz and Lellouch for trials where the purpose is to support real world clinical or policy decision making; and, on the other end of the spectrum, the overlap between the "Hypothetical strategy" as per ICH E9 R with the Explanatory approach suggested by S&L for trials whose research question focusses on confirming (or not), a	We agree that there is a degree of overlap, and this has been reflected in the text. We have also cited the relevant literature.

	causal hypothesis on the mechanism of action.	
3	I regret to say that I remain skeptical about the convenience of designing a pivotal trial based on these methods. Of course, I remain convinced that they provide a lot of clarity in the very difficult situation in which, due or to poor design or execution of the clinical trial, the treatment effect cannot be easily interpreted. By moving away from its experimental design, it is closer to the observational one and all interpretations are tentative and therefore cannot offer guarantees to future patients. In my opinion, estimands help to clarify how complicated it is to interpret a clinical trial in which, for some reason, decision-making is authorized once treatment has started, making untenable the assumption of a similar effect on which the authorization of the product rests for all patients within the eligibility criteria	We agree that the estimands framework is not a panacea. However, in our experience it does focus attention on the planning of the study, in particular what question the researchers want to answer. It also does make researchers think more about the impact of post-randomisation events and how these can be handled either through study conduct (i.e., educational support to patients and investigators) and how reducing missing data.
3	The authors speak of statistical significance 5 years later of the ASA statement. I interpret that only the P-value obtained from the main analysis defined in the protocol can be interpreted as a Neyman-Pearson test. The other P-values, without a priori power, should be eliminated.	The p-values all have exploratory value and are used to illustrate how the outcome can change under the varying strategies for handling post-randomisation events.